# Prelimbic Cortical Stimulation Induces Antidepressant-like Responses through Dopaminergic-Dependent and -Independent Mechanisms

**DOI:** 10.3390/cells12111449

**Published:** 2023-05-23

**Authors:** Sharafuddin Khairuddin, Wei Ling Lim, Luca Aquili, Ka Chun Tsui, Anna Chung-Kwan Tse, Shehani Jayalath, Ruhani Varma, Trevor Sharp, Abdelhamid Benazzouz, Harry Steinbusch, Arjan Blokland, Yasin Temel, Lee Wei Lim

**Affiliations:** 1Neuromodulation Laboratory, School of Biomedical Sciences, Li Ka Shing Faculty of Medicine, The University of Hong Kong, Hong Kong, China; 2Department of Biological Sciences, Sunway University, Bandar Sunway, Petaling Jaya 47500, Malaysia; 3Discipline of Psychology, College of Health and Education, Murdoch University, Perth 6150, Australia; 4Department of Pharmacology, University of Oxford, Oxford OX1 3QT, UK; 5CNRS UMR5293, Institute of Neurodegenerative Diseases, University de Bordeaux, 33000 Bordeaux, France; 6Department of Neuroscience, Maastricht University, 6229 ER Maastricht, The Netherlands; 7Department of Neuropsychology and Psychopharmacology, Faculty of Psychology and Neuroscience, Maastricht University, 6229 ER Maastricht, The Netherlands; 8Department of Neurosurgery, Maastricht University, 6229 HX Maastricht, The Netherlands

**Keywords:** high-frequency stimulation, depression, dopamine, susceptibility, resilience

## Abstract

High-frequency stimulation (HFS) is a promising therapy for patients with depression. However, the mechanisms underlying the HFS-induced antidepressant-like effects on susceptibility and resilience to depressive-like behaviors remain obscure. Given that dopaminergic neurotransmission has been found to be disrupted in depression, we investigated the dopamine(DA)-dependent mechanism of the antidepressant-like effects of HFS of the prelimbic cortex (HFS PrL). We performed HFS PrL in a rat model of mild chronic unpredictable stress (CUS) together with 6-hydroxydopamine lesioning in the dorsal raphe nucleus (DRN) and ventral tegmental area (VTA). Animals were assessed for anxiety, anhedonia, and behavioral despair. We also examined levels of corticosterone, hippocampal neurotransmitters, neuroplasticity-related proteins, and morphological changes in dopaminergic neurons. We found 54.3% of CUS animals exhibited decreased sucrose consumption and were designated as CUS-susceptible, while the others were designated CUS-resilient. HFS PrL in both the CUS-susceptible and CUS-resilient animals significantly increased hedonia, reduced anxiety, decreased forced swim immobility, enhanced hippocampal DA and serotonin levels, and reduced corticosterone levels when compared with the respective sham groups. The hedonic-like effects were abolished in both DRN- and VTA-lesioned groups, suggesting the effects of HFS PrL are DA-dependent. Interestingly, VTA-lesioned sham animals had increased anxiety and forced swim immobility, which was reversed by HFS PrL. The VTA-lesioned HFS PrL animals also had elevated DA levels, and reduced p-p38 MAPK and NF-κB levels when compared to VTA-lesioned sham animals. These findings suggest that HFS PrL in stressed animals leads to profound antidepressant-like responses possibly through both DA-dependent and -independent mechanisms.

## 1. Introduction

Depression is a common neuropsychiatric disorder that can often be very debilitating. Although symptoms of depression can vary between individuals, patients generally experience depressed mood and anhedonia [1]. A widely used animal model of depression is the chronic unpredictable stress (CUS) model, which has been used in preclinical studies to understand the pathophysiology of depression and to evaluate the pharmacological effects of antidepressants [2,3]. In this model, prolonged exposure to various unpredictable stressors in rodents induces anhedonia and depressive-like behaviors [2,4]. These features resemble the core symptoms in patients with major depression. However, a subpopulation of adult rodents does not exhibit anhedonic-like behaviors when exposed to different stress paradigms, suggesting such preclinical models of depression exhibit several phenotypes including resilience to stress and susceptibility to depression [5,6,7].

Human and animal studies have shown that the dopaminergic system in the brain reward circuit mediates stress responses, and that dysfunction of dopamine (DA) neurotransmission is associated with the pathogenesis and treatment of depression [8,9,10]. The midbrain DA neurons are known to exhibit heterogeneity in their afferent inputs and outputs and electrophysiological and functional properties depending on the specific projections to cortical and subcortical structures [11,12]. The inhibition of the ventral tegmental area (VTA) DA neurons projecting to the forebrain was found to modulate depressive-like behavior in the CUS model [13]. Several studies also reported that the activation and inhibition of VTA DA neurons mediated behavioral susceptibility and resilience through the VTA-nucleus accumbens or the VTA-medial prefrontal cortex (mPFC) circuit in mice in a social defeat stress model of depression [6,14,15,16]. On the other hand, a study proposed that the less-studied dorsal raphe nucleus (DRN) DA neurons have a role in mediating behavioral response to social isolation stress, which was also shown to be a distinct functional circuit from that of VTA DA neurons [17]. Taken together, these studies suggest an emerging role of the dopaminergic system in modulating stress responses through different functional groups of DA neurons and specific circuit pathways. 

Electrical high-frequency stimulation (HFS) has been gaining considerable attention as a potential treatment for neuropsychiatric and mood-related disorders [18,19,20]. Studies have shown that the application of HFS in different brain regions, such as the human subcallosal cingulate gyrus or the rodent homolog mPFC, can induce antidepressant effects [18,19]. We also showed that HFS of the mPFC (HFS mPFC) reduced depressive-like behaviors including anxiety, anhedonia, and forced swim immobility in a CUS animal model of depression [2]. Interestingly, these antidepressant effects were associated with changes in the neuronal activity of DRN, as demonstrated by altered serotonin (5-HT) neurotransmission and c-Fos expression [2]. These results imply that HFS mPFC potentially modulates changes in the neurocircuitry of mood-related disorders. However, the effects of DA regulation and HFS specifically in the prelimbic cortex (PrL) of the mPFC are still unknown.

In this study, we investigated the effects of HFS of the prelimbic cortex (HFS PrL) on stress susceptibility and resilience to various depressive-like behaviors in an animal model of depression. The delineation of stress susceptibility and resilience phenotypes was based on the animal’s stress response to sucrose intake during the CUS paradigm. Anxiety-like responses, anhedonia, and despair-like behavioral tests were also performed to assess the antidepressant-like effects of HFS PrL. To further study the effects of HFS PrL on the susceptibility and resilience to depression in the CUS rat model, we examined the level of stress hormones, hippocampal monoaminergic neurotransmitters, and morphological changes in midbrain DA neurons. We also examined the role of DA mechanisms in modulating the antidepressant-like effects of HFS PrL via bilateral 6-hydroxydopamine (6-OHDA)-induced lesioning of DA neurons in the DRN and VTA.

## 2. Materials and Methods

### 2.1. Subjects

Twelve-week-old adult male Sprague-Dawley rats (300–400 g; *n* = 153) were individually housed with food and water *ad libitum*. The holding room was maintained under a 12 h dark-light cycle (lights off at 09:00) in a controlled temperature (21 ± 1 °C) and humidity (60–70%) environment. All experimental procedures were approved by the Committee on the Use of Live Animals in Teaching and Research (CULATR No. 3946-16) at the University of Hong Kong.

### 2.2. Chronic Unpredictable Stress (CUS) Animal Model of Depression

The CUS paradigm was performed as previously described by our laboratory [2,21]. The CUS protocol consisted of continuous exposure to various stressors for 10–14 h per day for 5 weeks. The stressors included stroboscopic light flashes (2.5 Hz), intermittent illumination (alternate 2 h on/off), housing in mouse cages, soiled-cage with 300 mL cold water, paired-housing in dirty cages (with excreta of another rat), food and water deprivation, and no stressors. The order of stressors was randomized to maintain the nature of the unpredictable stress, with one stressor carried out in the morning and another in the evening. The CUS and non-CUS control groups were placed in separate animal holding rooms to avoid cross-exposure. The non-CUS controls were handled by researchers daily to avoid human stress during behavioral testing. After 21 days of CUS, animals were subjected to electrical stimulation (1 h daily HFS for two weeks; from CUS day 22–36) and behavioral testing to assess anxiety, anhedonia, behavioral despair or antidepressant-like activities. Behavioral testing was conducted on specific days in weeks 4 and 5 of CUS (i.e., CUS day 20–35). Stressors were applied continuously throughout the entire experiment except during the behavioral testing. 

### 2.3. Surgical Procedures and Deep Brain Stimulation

The procedures for surgery and electrical stimulation were conducted as previously described by our laboratory [2,22,23,24]. Rats in stereotactic apparatus (Vernier Stereotaxic Instrument, Leica Biosystems, Nussloch, Germany) were anesthetized by 2.5% isoflurane inhalation. After making burr holes, stimulating electrodes were implanted in the PrL (AP: +3.24 mm, ML: 0.7 mm, VL: −3.0 mm) according to coordinates derived from the brain atlas of Paxinos and Watson [25]. The electrodes (Synergy Pte Ltd., Singapore) consisting of a gold-plated needle-like electrode with a platinum–iridium inner wire (Technomed, Beek, The Netherlands) were secured using dental cement (Heraeus Kulzer, Hanau, Germany). In the post-surgical period, all animals were treated with enrofloxacin (5 mg/kg) and buprenorphine hydrochloride (0.3 mg/kg) for 7 days as recommended by CULATR. For electrical stimulation, each electrode was connected to a MultiStim-8 channel stimulator with isolators (Model 3800 and Model 3820; A-M Systems, Washington, United States). The HFS PrL stimulation parameters were a frequency of 100 Hz, amplitude at 100 μA and pulse width of 100 μs (charge-balanced biphasic/bipolar pulses), which were based on our previous study that demonstrated these parameters had profound antidepressant-like effects [2,21]. The sham or control rats were connected to the stimulation cables without any electrical stimulation. 

### 2.4. Experiment 1: HFS PrL in CUS-Susceptible and -Resilient Models

In experiment 1 (Figure 1), the procedures for stereotaxic surgery, CUS paradigm, deep brain stimulation, and behavioral testing were conducted as previously described [2,21,22,24]. After 2 weeks of recovery from the electrode implantation procedure, animals were subjected to the CUS paradigm for 5 weeks. Blood samples were collected on the day before the CUS procedure and after 1 h of HFS PrL on the day of sacrifice. The sucrose intake test was conducted after 3 weeks of CUS (on day 20–22) and at week 4 after HFS PrL (on day 30–32). Animals were categorized as either susceptible or resilient to CUS-induced depression based on the sucrose intake test after week 3 of CUS. In accordance with earlier studies, animals were categorized as susceptible to depressive-like or anhedonia-like behavior if there was a 40% reduction in sucrose consumption compared with the average sucrose intake of the non-CUS controls [5,26], whereas the remaining animals were categorized as resilient to depressive-like behavior (Figure 1B,C).

The animals were assigned into groups as follows: non-CUS control (*n* = 15), CUS sham (*n* = 28), and CUS HFS PrL (*n* = 28). Animals that had electrode detachment (CUS-sham, *n* = 2; CUS-HFS PrL, *n* = 1), misplacement of electrodes (CUS-HFS PrL, *n* = 2), died during/after surgery (non-CUS control, *n* = 1; CUS-sham, *n* = 2; CUS-HFS PrL, *n* = 1), or were severely sick during the experiments (non-CUS control, *n* = 1; CUS-sham, *n* = 2) were excluded from the study. The final number of animals per group were as follows: non-CUS control, *n* = 13; CUS-susceptible sham, *n* = 12; CUS-susceptible HFS PrL, *n* = 13; CUS-resilient sham, *n* = 10; and CUS-resilient HFS PrL, *n* = 11. The electrode localizations were confirmed to be within the PrL in the range of Bregma level from 4.2 to 3.0 mm (Figure 1D). 

On day 24 of CUS, animals underwent the home cage emergence test to assess levels of anxiety [2]. On days 30–32 of CUS, animals were subjected to the sucrose intake test to evaluate changes in hedonia. Finally, on days 34–35 of CUS, animals underwent the forced swim test to assess behavioral despair and antidepressant-like responses. In all behavioral tasks, animals were stimulated for 15 min prior to each test with continual stimulation during the entire testing phase, except the forced swim test received stimulation only prior to testing. Sham and non-CUS control groups were similarly treated, but did not receive any electrical stimulation. The behavioral analyses were performed using ANY-maze 5.0 software by experienced researchers who were blinded to the experimental conditions as previously described [2]. After the behavioral experiments, animals were sacrificed either by immediate decapitation for assessing changes in hippocampal neurotransmission by mass spectrometry (Figure 2 and Figure 3, *n* = 4–5 animals in each group) or by intracardial perfusion of 4% paraformaldehyde for immunohistochemical study (Figure 4; *n* = 5–7 animals in each group). 

### 2.5. Experiment 2: HFS PrL in Animal Models with 6-OHDA Lesioning in DRN and VTA

In Experiment 2 (Figure 5), to distinguish the underlying mechanisms of the role of DA-dependent and -independent processes on the antidepressant-like effects by HFS PrL, dopaminergic cells in the DRN (AP: −7.4 mm, ML: +1.0 mm, VL: −5.8 mm; approached at a coronal angle of 10°) and VTA (AP: −5 mm, ML: ±0.8 mm, VL: −8 mm) were lesioned by 6-OHDA, a potent neurotoxin for dopaminergic neurons [27]. Animals were placed on a stereotactic apparatus and anesthetized with 2.5% isoflurane throughout the surgical procedure. The 6-OHDA solution was prepared by dissolving 4 µg/µL of 6-Hydroxydopamine hydrochloride (ChemCruz, California, CA, USA.) in 0.2% ascorbic acid in 0.9% saline. Lesioning in the DRN or VTA was performed by micro-injection of 2 µL 6-OHDA solution at an injection rate of 0.1 µL per min through a stainless-steel injection cannula (33 gauge; C315FDS-4/SPC, Plastics One Inc., Roanoke, VA, USA) connected by tubing to a 10 µL Hamilton syringe and infusion pump (Pump11 Elite, Harvard Apparatus, Holliston, MA, USA). After injection, the needle was left in place for 15 min to allow the 6-OHDA to completely diffuse within the injected area. After lesioning, stimulating electrodes were implanted in the PrL (AP: +3.24 mm, ML: +/−0.7 mm, VL: −3.0 mm) and animals were allowed to recover for 2 weeks. To protect the noradrenergic neurons from lesioning by 6-OHDA, a single dose of desipramine hydrochloride (25 mg/kg in 0.9% NaCl solution; Sigma-Aldrich, St. Louis, MO, USA) was injected intraperitoneally 30–60 min prior to the surgery [28]. 

Animals were initially assigned to the groups as follows: non-lesioned control, *n* = 14; DRN-lesioned sham, *n* = 12; DRN-lesioned HFS PrL, *n* = 12; VTA-lesioned sham, *n* = 12; and VTA-lesioned HFS PrL, *n* = 12. Animals with electrode detachment/misplacement (DRN-lesioned HFS PrL, *n* = 1; VTA-lesioned sham, *n* = 1 and HFS PrL, *n* = 1) and severely sick or dead due to 6-OHDA injection (non-lesioned control, *n* = 2; DRN-lesioned sham, *n* = 2; DRN-lesioned HFS PrL, *n* = 1; VTA-lesioned sham, *n* = 2; and VTA-lesioned HFS PrL, *n* = 2) were excluded from the study. The final number of animals in each group were as follows: non-lesioned control, *n* = 12; DRN-lesioned sham, *n* = 10; DRN-lesioned HFS PrL, *n* = 10; VTA-lesioned sham, *n* = 9; and VTA-lesioned HFS PrL, *n* = 9. Animals were behaviorally assessed in the home cage emergence test (on day 21), sucrose intake test (on day 24), and forced swim test (on days 27–28). Animals received 1 h of daily stimulation for 2 weeks (i.e., days 15–29). For behavioral testing, animals were stimulated for 15 min prior to each test with continual stimulation throughout the entire testing period, except the forced swim test where animals received stimulation only prior to testing. The sham and control groups were similarly treated, but they did not receive any electrical stimulation. After behavioral testing, animals were sacrificed by either intracardial perfusion of 4% paraformaldehyde for immunohistochemical study (Figure 5; *n* = 4–5 animals in each group) or by immediate decapitation for neurotransmitter assessment by mass spectrometry (Figure 6, *n* = 4–5 animals per pool, in triplicate) and hippocampal protein expression assessment by Western blotting (Figure 7; *n* = 4–6 animals in each group). All electrode localizations were verified to be within the PrL region in the range of the Bregma level between 4.2 and 3.0 mm (Figure 5B). 

### 2.6. Behavioral Experiments

Home cage emergence test: the test was conducted in a home cage with the lid removed and a grid placed over the edge of the home cage to allow the animal to escape. The home cage and a new cage, each measuring 42.5 × 26.6 × 18.5 cm, were placed next to each other with the wire grid ramp connecting the two. The duration of escape latency was measured during a 10 min trial [2,29,30,31]. 

Sucrose intake test: animals were habituated to drinking 1% sucrose solution for 1 h on the day prior to testing. The HFS PrL or sham animals were subjected to 14 h (20:00–10:00 next day) of fasting and were deprived of food and water. Subsequently, all animals were exposed to 1% sucrose solution for 1 h at 10:00–11:00. The sucrose intake level was calculated as the total amount of sucrose solution consumed normalized by body weight (g/kg), as previously described [2]. 

Forced swim test: the test was conducted in a transparent Perspex cylinder (50 × 20 cm) with tap water (25 ± 1 °C) filled to a depth of 30 cm [2,31]. The test was carried out on 2 consecutive days. All animals were habituated in the cylinder of water for a period of 15 min. On the next day, animals were stimulated for 15 min and then tested in the cylinder of water for 10 min. The duration of immobility was analyzed by researchers who were blinded to the experimental conditions. 

### 2.7. Immunohistochemistry

One day after the behavioral study, all rats received 1 h of stimulation. Animals immediately received an injection of sodium pentobarbital (Dorminal 20%, Alfasan, Woerden, Holland) and then intracardially perfused with 4% paraformaldehyde fixative solution. Brains were removed and serially cut into 30 μm coronal sections and stored at −80°C. Histological staining for tyrosine hydroxylase-immunoreactive (TH-ir) cells was carried out as previously described [32]. In brief, brain sections were incubated with rabbit anti-tyrosine hydroxylase (ab112, 1:1000; Abcam, Cambridge, MA, USA) overnight at 4 °C with constant shaking. After rinsing, all sections were incubated with a biotinylated anti-rabbit IgG secondary antibody (1:500; Vector Laboratories, Inc, Burlingame, CA, USA) for 90 min. The sections were incubated with an avidin-biotin-peroxidase complex (1:1000, Vectastain Elite, Vector Laboratories, Newark, CA, USA) for 120 min. The sections were rinsed and subsequently incubated in a solution of 3,3′-diaminobenzidine tetrahydrochloride (DAB Substrate Kit; Sigma-Aldrich, St. Louis, MO, USA) with nickel chloride to enhance the visualization of the horseradish-peroxidase reaction product. Lastly, sections were mounted on gelatin-coated slides, dehydrated, and cover-slipped with Permount^TM^ (Thermo Fisher Scientific Inc., Waltham, MA, USA). The quantification of TH-ir cells was performed in the VTA (Bregma level: from −5.0 to −6.0 mm) and different regions of the DRN (including dorsal raphe dorsal, dorsal raphe ventral, dorsal raphe ventrolateral, and median raphe nucleus; Bregma level: from −7.3 to −8.2 mm), as previously described [32]. Photomicrographs of TH-ir cells within the regions of interest (4–5 sections per animal) were taken using an Olympus DP73 digital camera (Olympus, Hamburg, Germany) attached to an Axiophat 2 imaging microscope (Carl Zeiss Microscopy GmbH, Gottingen, Germany) and quantification was performed using ‘Image J’ (version 1.38, NIH, Bethesda, MD, USA) as previously described [32]. In-section artefacts were excluded from the analysis to ensure accuracy of the measurements. 

### 2.8. Corticosterone Radioimmunoassay 

Blood samples were collected from rat tail vein directly in ice-cold heparinized capillary tubes (Microvette, CB300, Sarstedt, Germany) at 4 °C. After centrifugation at 3000 rpm for 5 min at 4 °C, plasma was extracted with 3 mL dichloromethane. For measuring CORT, 1 mL of the extract was dried for radioimmunoassay using CORT−^125^I. The radioimmunological reaction was performed overnight at 4 °C, followed by a secondary antibody system to separate bound and unbound steroids. The radioimmunoassay procedure and measurement of CORT was performed as previously described [33,34].

### 2.9. Mass Spectrometry

The mass spectrometry analysis was conducted as previously described by our laboratory [23,24,30]. All animals from experiments 1 and 2 were deeply anesthetized with pentobarbital (200 mg/kg) before decapitation. After sacrifice, brains were extracted and immediately frozen in liquid nitrogen and stored at −80 °C. The hippocampal regions were cryo-sectioned at a thickness of 100 μm using a cryostat (Leica CM3050S, Nussloch, Germany), and a total of 400 μm of hippocampal slices were used. For mass spectrometry (Figure 2 and Figure 5), hippocampal sections in 1.5 mL Eppendorf tubes were processed accordingly. Levels of DA, homovanillic acid (HVA), dihydroxyphenylacetic acid (DOPAC), norepinephrine, 5-HT, 5-Hydroxyindole acetic acid (5-HIAA), glutamate, and gamma-aminobutyric acid (GABA) were measured using norvaline as the internal standard by GC-MS on an Agilent 7890B GC and Agilent 7010 Triple Quadrupole Mass Spectrometer system (Agilent, CA, USA). Characteristic quantifier and qualifier transitions were monitored in MRM mode and spectra from m/z 50–500 were acquired in SCAN mode. Data analysis was performed using the Agilent MassHunter Workstation Quantitative Analysis Software (https://www.agilent.com/zh-cn/product/software-informatics/mass-spectrometry-software/data-analysis/quantitative-analysis, accessed on 17 May 2023). Linear calibration curves for each analyte were generated by plotting the peak area ratio of external/internal standard against the standard concentration at different concentration levels. Analytes were confirmed by comparing the retention time and ratio of characteristic transitions between the sample and standard. 

### 2.10. Western Blot Analysis 

The western blot experiments were performed as previously described by our laboratory [30,31]. The hippocampus was micro-dissected and a total of 400 μm of hippocampal slices were used for the western blotting. Samples were homogenized with RIPA buffer containing protease and phosphatase inhibitors (Thermo Scientific, Rockford, IL, USA). The protein concentration was measured by Bio-Rad DC Protein Assay Kit (Bio-Rad, Hercules, CA, USA). Each sample was separated by 8–12% SDS-PAGE and transferred to a PVDF membrane (Bio-Rad Laboratories, Hercules, CA, USA) using a semi-dry electroblotting system. The membranes were blocked with 5% BSA in TBS-T (20 mM Tris-HCl, 150 mM NaCl, 0.1% Tween 20) for 1 h at room temperature. Blots were incubated at 4 °C overnight with the respective primary antibodies (1:1000 dilution; Cell Signaling Technology, Inc., Beverly, MA, USA; unless otherwise indicated). Antibodies included Akt/protein kinase B (Akt), phosphorylated-Akt (p-Akt), extracellular signal-regulated kinase 1/2 (Erk1/2), phosphorylated Erk (p-Erk1/2), postsynaptic density protein 95 (PSD-95; 1:1000 dilution; Abcam, Cambridge, MA, USA), glial fibrillary acidic protein (GFAP), p38 group of mitogen-activated protein kinases (p-38 MAPK), phosphorylated p38 (p-p38 MAPK), nuclear factor κB p65 (D14E12) XP^®^ (NF-κB), protein kinase A C-α (PKA), phosphorylated PKA C^Thr197^ (p-PKA), phosphorylated glycogen synthase kinase 3 beta (p-GSK-3β), caspase-3, and glyceraldehyde 3-phosphate dehydrogenase (GAPDH). Horseradish peroxidase-conjugated anti-rabbit or anti-mouse immunoglobulin G antibody (Invitrogen, Thermo Fisher Scientific, Waltham, MA, USA) was added for 1 h at room temperature, followed by visualization using a chemiluminescence kit (Bio-Rad Laboratories, Inc., Hercules, CA, USA). The relative protein expression was normalized against GAPDH. 

### 2.11. Statistical Analysis

Data analysis was performed using IBM SPSS Statistics and all results were presented in bar or line graphs as mean ± S.E.M (individual data points). The Shapiro-Wilk test was used to test the normality of the results. The non-parametric test (Kruskal-Wallis) was used to analyze non-normally distributed data as appropriate. A one-way or two-way ANOVA followed by Tukey’s HSD post-hoc test for multiple comparisons were used to analyze parametric data as appropriate. An independent sample *t*-test or Mann-Whitney U test was used to analyze the difference between control sham and control HFS PrL. Square root transformation was applied to the plasma CORT data, followed by repeated measures ANOVA. All *p*-values < 0.05 were considered significant.

## 3. Results

### 3.1. CUS Exposure Induces Stress Susceptibility and Resilience to Depression 

We found that exposure to stressors in the CUS paradigm (Figure 1A) induced a significant reduction in sucrose intake in 54.3% of CUS animals (*n* = 25 out of a total 46 CUS animals) who were assigned to the CUS-susceptible group, whereas the remaining animals were assigned to the CUS-resilient group (Figure 1B,C). The CUS-susceptible and CUS-resilient groups were randomly assigned to either HFS PrL (*n* = 24) or sham (*n* = 22) and their antidepressant-like behavior was then assessed. Comparing between CUS-susceptible sham and non-CUS control groups, we found the CUS-susceptible sham animals had significantly increased escape latency in the home cage emergence test (*p* = 0.001; Figure 1E), reduced sucrose consumption (*p* = 0.010; Figure 1F), and increased forced swim immobility (*p* < 0.001; Figure 1G), indicating the successful induction of depressive-like behaviors including anxiety, anhedonia, and behavioral despair. In the home cage emergence test, HFS PrL reduced escape latency in CUS-susceptible animals compared with CUS-susceptible sham animals (*p* = 0.002; Figure 1E). In the sucrose intake test, the HFS PrL groups showed increased sucrose consumption compared with CUS-susceptible or -resilient sham groups, respectively (*p* < 0.003; Figure 1F). In the forced swim test, there were significant differences in both CUS-susceptible and -resilient sham groups compared with non-CUS controls (*p* < 0.001). In line with previous studies [2,4], HFS PrL significantly reduced the duration of immobility to a greater extent compared with CUS-susceptible and -resilient sham groups, respectively (*p* < 0.001; Figure 1G). We further investigated the effects of HFS PrL in control non-CUS animals, which showed no significant differences in the sucrose intake test at week 3 (before HFS), sucrose intake test at week 4 (after HFS), home cage emergence test, and forced swim test (all *p* < 0.461) compared with non-CUS control sham animals (Figure 3A–D). 

### 3.2. HFS PrL in CUS-Resilient Animals Decreases Corticosterone Levels and Increases DA Levels and the Number of Midbrain DA Cells

There were no significant differences in corticosterone (CORT) levels during pre-CUS baseline measurements (Figure 2A); however, we observed increased plasma CORT levels in CUS-susceptible sham and HFS PrL, and CUS-resilient sham animals after week 5 of the CUS treatment compared with their respective baseline levels and with non-CUS controls (*p* < 0.001; Figure 2A). Repeated measures ANOVA demonstrated significant main effects of time (F_(1,53)_ = 36.174, *p* < 0.001), group (F_(4,53)_ = 12.038, *p* < 0.001), and interactions of time x group (F_(4,53)_ = 10.667, *p* < 0.001). On day 36 of CUS before animals were sacrifice, HFS PrL in both CUS-susceptible and -resilient groups significantly reduced CORT levels compared to their respective CUS sham groups (*p* < 0.044). In non-CUS control groups, we did not observe significant effects of time (F_(1,20)_ = 0.653, *p* = 0.429), group (F_(1,20)_ = 0.209, *p* = 0.652), and their interactions (F_(1,20)_ = 0.902, *p* = 0.534) on plasma CORT levels between non-CUS control sham and HFS PrL groups (Figure 3E).

Both CUS-susceptible and -resilient sham groups had decreased DA levels compared with non-CUS controls (*p* < 0.001; Figure 2B), whereas HFS PrL increased DA levels in both CUS-susceptible and -resilient groups compared with their respective sham groups, (*p* < 0.001). The CUS-susceptible sham group also showed reduced levels of hippocampal 5-HT compared with non-CUS controls (*p* = 0.024; Figure 2C). We found HFS PrL in both CUS-susceptible and -resilient groups enhanced the level of 5-HT compared with their respective sham groups (*p* < 0.001; Figure 2C), which was supported by previous studies that showed HFS of the mPFC increased levels of hippocampal 5-HT [4,24]. There were no changes in the levels of HVA, DOPAC, 5-HIAA, norepinephrine, glutamate, and GABA among the groups (Figure 2B,C). We observed no significant changes in DA, HVA, DOPAC, norepinephrine, 5-HT, 5-HIAA, Glutamate, and GABA (all *p* < 0.805) between non-CUS control HFS PrL animals and non-CUS control sham animals (Figure 3F,G).

Studies have shown that a reduction in midbrain DA neurons in patients with Parkinson’s disease is associated with anhedonia and loss of motivation [35]. Although we did not find significant differences in the number of tyrosine hydroxylase-immunoreactive (TH-ir) cells in the VTA and DRN between CUS-susceptible sham and non-CUS animals (Figure 4), HFS PrL in both CUS-susceptible and -resilient animals increased the number of TH-ir cells in the dorsomedial part of the DRN compared with non-CUS control animals (*p* < 0.017; Figure 4B). There was a significant increase in TH-ir cell count in the DRN of CUS-resilient HFS PrL animals compared with both CUS-susceptible and -resilient sham animals (*p* < 0.045). Additionally, we also detected increased TH-ir cell counts in the VTA region of both CUS-resilient sham and HFS PrL groups compared with CUS-susceptible sham and non-CUS control groups (*p* < 0.018; Figure 4A). No significant differences were found in TH-ir cell counts in the dorsal raphe ventral, dorsal raphe ventrolateral, and median raphe nucleus among the groups (Figure 4C–E).

### 3.3. Effects of HFS PrL on DA Lesioning in the DRN and VTA 

In non-lesioned control groups, we found no significant changes between HFS PrL and sham groups in the home cage emergence test, sucrose intake test, and forced swim test (all *p* < 0.881; Figure 5C–E). In the home cage emergence test, we found no significant differences in the escape latency of DRN-lesioned HFS PrL animals compared with DRN-lesioned sham animals (Figure 5F). In VTA-lesioned HFS PrL animals, we observed reduced escape latency when compared with the VTA-lesioned sham group (*p* = 0.035). The DRN-lesioned sham and HFS PrL groups and the VTA-lesioned sham group showed decreased sucrose consumption (*p* < 0.022), but not in the VTA-lesioned HFS PrL group when compared with non-lesioned controls (Figure 5G). Although no differences were demonstrated in the forced swim immobility of DRN-lesioned sham and HFS PrL groups, we observed an increase in immobility time in the VTA-lesioned sham group compared with the non-lesioned controls (*p* = 0.002). Interestingly, this immobility behavior was completely reversed in VTA-lesioned HFS PrL animals (*p* = 0.011; Figure 5H). In DRN-lesioned animals, there were reduced TH-ir cell counts in both sham (48.8%) and HFS PrL (46.3%) groups compared with non-lesioned controls, respectively (*p* < 0.003; Figure 5I,J,L). Meanwhile, VTA-lesioned animals showed a 34.7% and 40.9% decrease in TH-ir cell count in sham and HFS PrL groups compared with non-lesioned controls (*p* < 0.001; Figure 5I,K,L), respectively. 

### 3.4. HFS PrL Enhances Hippocampal DA Neurotransmission and Neuroplasticity-Related Protein Expression in VTA-Lesioned Animals 

There were significant decreases in DA levels in DRN-lesioned sham and HFS PrL groups, and in the VTA-lesioned sham group compared with non-lesioned controls (*p* < 0.001; Figure 6A). We observed the VTA-lesioned HFS PrL group had increased DA levels (*p* < 0.001) and decreased norepinephrine levels (*p* = 0.009) compared with the sham group. Moreover, both VTA-lesioned sham and HFS PrL animals had decreased DOPAC levels compared with non-lesioned controls (*p* < 0.043). Although there were no changes in 5-HT level, VTA-lesioned HFS PrL animals showed decreased 5-HIAA level (*p* = 0.045). No differences were found in levels of HVA, glutamate, or GABA among the groups (Figure 6).

Examining the DA-independent mechanisms of the antidepressant-like effects of HFS PrL, we found increased hippocampal protein expressions of p-Akt (*p* = 0.003) and PSD-95 (*p* = 0.026) in VTA-lesioned HFS PrL animals compared with VTA-lesioned sham animals (Figure 7B,F). Although no differences in the protein expression of GFAP was observed between DRN-lesioned sham and HFS PrL groups, we found a significant increase in GFAP protein level in the VTA-lesioned HFS PrL group compared with the VTA-lesioned sham group (*p* = 0.034; Figure 7G). We also observed remarkable increases in Erk1/2 protein levels in the DRN-lesioned and VTA-lesioned sham groups compared with non-lesioned controls (*p* < 0.001). Interestingly, HFS PrL normalized the Erk1/2 protein expression back to the level in non-lesioned controls (*p* < 0.005; Figure 7E). We also found the protein expression levels of p-p38 MAPK and NF-κB were remarkably increased in VTA-lesioned sham animals compared with non-lesioned controls (*p* < 0.033), whereas VTA-lesioned HFS PrL animals showed suppressed p-p38 MAPK and NF-κB signaling compared with VTA-lesioned sham animals (*p* < 0.026; Figure 7H,J). No significant differences were found in the protein expression of Akt, p-Erk1/2, p38 MAPK, PKA, p-PKA, p-GSK-3β, and caspase-3 among the groups (Figure 7).

## 4. Discussion

Maladaptation to chronic stress is associated with the development of depression, which often manifests as adversely altered psychological and neurophysiological responses [1,36]. However, not all individuals exposed to chronic stress will develop major depression, and a subset of the human population are actually resilient to stress [37]. This resilience allows these individuals to adapt to chronic stress conditions with minimal disturbance to their emotional wellbeing and prevents the development of psychopathology [37]. Nevertheless, the neurobiological mechanisms underlying resilience in such individuals remain largely unknown. Our results revealed that 54.3% of animals exposed to CUS developed susceptibility to depression compared with non-CUS controls according to an operational cut-off of a 40% reduction in sucrose consumption. This result is in line with previous studies by Bergström et al. which demonstrated the use of the sucrose intake test as a reliable method to measure the level of an anhedonic phenotype in a CUS animal model of depression for assigning animals to the CUS-susceptible and CUS-resilient groups [5,26]. Additionally, we showed there was an increase in the level of CORT stress hormone in CUS-susceptible and -resilient sham groups compared with the levels at baseline and with non-CUS controls. The hypothalamic–pituitary–adrenal axis receives various afferent projections from the limbic areas including the prefrontal cortex, nucleus accumbens, hippocampus, and amygdala [38]. Rodents that are stress susceptible to depression commonly show increased plasma CORT levels [7] and can also have abnormal dopamine functions. Studies have found that animals exposed to a social defeat paradigm showed long-lasting changes in dopamine levels, such as decreased basal dopamine, reduced D2 receptor expression, enhanced monoamine oxidase A gene expression, and DA transporter binding in the prefrontal cortex [39,40]. Our present study demonstrated that HFS PrL decreased CORT levels and enhanced DA levels in the hippocampus of both CUS-susceptible and -resilient groups compared with their respective sham groups.

We conducted various behavioral tests on the CUS rat model to assess anxiety, anhedonia, and behavioral despair, which showed this animal model mimicked various aspects of the clinical symptoms of major depression [1,2]. Compared with non-CUS controls, we observed that CUS-susceptible sham animals displayed anxiety-like behavior in the home cage emergence test, anhedonic behavior in the sucrose intake test, and behavioral despair in the forced swim test. Previous studies demonstrated that HFS mPFC also induced antidepressant-like effects. In this study, we found that HFS PrL in both CUS-susceptible and -resilient groups induced anxiolytic effects in the home cage emergence test compared with CUS-susceptible sham animals. This observation possibly suggests that HFS PrL is effective in alleviating anxiety (e.g., generalized anxiety disorder or post-traumatic stress disorder). We observed that HFS PrL significantly enhanced hedonic-like responses in both CUS-susceptible and -resilient animals in the sucrose intake test. The hedonic-like effects were more pronounced in the CUS-resilient group, indicating the importance of stress resilience in reducing anhedonia symptoms in depression [41]. We found that HFS PrL in both CUS-susceptible and -resilient groups significantly reduced behavioral despair in the forced swim test compared with both sham groups, which is in line with previous studies that reported HFS-induced antidepressant effects [2,4,18].

Previous studies reported that 5-HT mediated the effects of HFS mPFC [2,4]. In this study, we found that HFS PrL not only enhanced 5-HT levels but also increased DA levels in both CUS-susceptible and -resilient animals compared with their respective sham groups. Interestingly, levels of 5-HT and DA neurotransmitters were greatly enhanced in CUS-resilient HFS PrL animals compared with CUS-susceptible HFS PrL animals, possibly suggesting a mechanism of resilience underlying the enhanced antidepressant-like effects in the sucrose intake and forced swim tests. We observed no changes in the levels of hippocampal HVA, DOPAC, 5-HIAA, glutamate, and GABA. In contrast, another study reported downregulated levels of 5-HT and HVA, and upregulated levels of norepinephrine and DOPAC in the hippocampus of a rodent model of post-traumatic stress disorder under predator exposure/psychosocial stress [42]. Our previous microarray and gene expression studies in aged animals showed that HFS PrL increased DA levels and upregulated Drd1, Drd2, and Htr1d in the hippocampus [24]. A study showed the interaction of DA with 5-HT was mainly through Drd2 receptors [43]. Importantly, the present findings agree with the previous research that showed stress induced reductions of 5-HT and DA in the hippocampus [44,45]. 

Of particular interest, studies have found that dysfunctions in the DA system are associated with anhedonia and disrupted responsiveness to conditioned incentive stimuli and motivational reward prediction [46,47]. Findings from animal models of stress-induced depressive-like behavior demonstrated there were abnormal changes in DA receptor expression in various structures of the mesolimbic system [48,49]. A recent study by Laudani et al. showed the associations between altered composition of gut microbiota and dopaminergic abnormalities can lead to trauma susceptibility in post-traumatic stress disorder [50]. They found increased levels of L-tyrosine-derived metabolite p-cresol and D3 receptor expression were associated with dysregulated levels of dopamine and DOPAC specifically in the prefrontal cortex of mice. Many human studies reported that depleting or blocking DA using pharmacological approaches induced depression [51]. Recently, a human study using pharmacological functional magnetic resonance imaging revealed that subjects with severe post-traumatic stress disorder receiving tolcapone (a drug that enhances cortical dopamine tone through inhibiting the degradation of dopamine by catechol-O-methyl transferase) showed significant improvements in working memory performance and affective dysfunction symptoms [52]. Taken together, these findings indicate dysfunctions in the mesolimbic DA system could play a possible causal role in the induction of stress and depressive-like behaviors as observed in patients and animal models.

In the DRN, we observed only the dorsomedial region had increased TH-ir cell counts in both CUS-susceptible and -resilient HFS PrL groups compared with non-CUS controls. Similarly, we observed the VTA had increased TH-ir cell counts in both CUS-resilient sham and HFS PrL groups compared with non-CUS controls and CUS-susceptible sham groups, indicating DA has an important role in counteracting the high-demand of DA neurotransmission during stress. There is abundant evidence showing that the projection of the prefrontal cortex to the VTA and its stimulation regulate DA neuronal activity and extracellular levels within forebrain regions [53,54]. Recent studies suggest that DA neurons in the DRN are involved in social interaction [17], reward memories [55], arousal, wakefulness [56], and fear response [57]. It was demonstrated that optogenetic phasic stimulation of VTA DA neurons projecting to the mPFC induced susceptibility to social-defeat stress as characterized by social avoidance and decreased sucrose preference [15]. Similarly, Tye et al. reported that optogenetic inhibition of VTA neurons instantly induced depressive-like behaviors as measured by increased tail-suspension immobility and anhedonia in the sucrose preference test [13]. 

Animals with 6-OHDA lesioning in the VTA had induced learned helplessness behavior [58]. In regard to anhedonia in Parkinson’s disease [59], decreased levels of DA were observed in both the substantia nigra and VTA regions in depressed patients with Lewy body disorders [59,60]. Animal models have shown that depressive-like symptoms are causally linked to hyperactivity of DA neurons in the VTA [15], and that exposure to chronic stress at weeks 2, 4, 8, and 16 induced 9.8%, 19.2%, 39.5%, and 40.6% TH-ir neuronal loss in the VTA, respectively, when compared with the controls [61]. This raises the question of whether the involvement of DA populations within the VTA and DRN similarly regulate the depressive-like behaviors induced by HFS PrL. To examine the DA-dependent mechanism of the antidepressant effects of HFS PrL, we performed 6-OHDA lesioning of neurons in the DRN and VTA. The DRN-lesioned HFS PrL animals showed increased anxiety in the home cage emergence test compared with non-lesioned controls, but no effects were found in the DRN-lesioned sham animals. In the sucrose intake test, DRN-lesioned sham and HFS PrL animals and VTA-lesioned sham animals had increased anhedonic-like responses, but not in VTA-lesioned HFS PrL animals when compared with non-lesioned controls. This possibly indicates that the hedonic-like effects of HFS PrL are dependent on a dopaminergic mechanism in the DRN and VTA. In the forced swim test, VTA-lesioned sham animals showed behavioral despair compared with non-lesioned controls, as demonstrated by increased forced swim immobility, whereas HFS PrL reduced the immobility time back to that of the non-lesioned controls. This indicates the antidepressant-like response of HFS PrL was through a DA-independent mechanism in the VTA. On the other hand, the antidepressant-like effect of HFS PrL in VTA-lesioned animals was possibly mediated through a DA-dependent mechanism in the DRN. Future studies involving lesioning of both the DRN and VTA regions could further delineate the overall DA-dependent and/or -independent mechanisms of the antidepressant-like responses by HFS PrL. 

The hippocampus is innervated by dopaminergic fibers that project from the VTA [62,63]. The mesolimbic system provides DA innervation to several subcortical regions including the nucleus accumbens, septum, olfactory tubercle, hippocampus, and amygdala. Moreover, DA was shown to modulate neuroplasticity in dentate granule cells in humans and in rodent models [64]. Although previous studies showed that HFS mPFC enhances neuroplasticity in the hippocampus [22,30,65], the underlying DA-dependent or -independent mechanisms of the antidepressant-like effects of HFS PrL in the hippocampus remain obscure. In this study, we found increased hippocampal protein expressions of p-Akt and PSD-95 in VTA-lesioned HFS PrL animals compared with VTA-lesioned sham animals, which possibly indicates the activation of p-Akt regulates the synaptic function of PSD-95 through vesicular transport [66,67]. Both animal and clinical studies on depression have shown there is decreased GFAP [68,69] together with increased neuroinflammation as seen by the expression of Erk1/2 [70,71], p-p38 MAPK [72], and NF-κB [73,74]. We showed there was increased GFAP and decreased Erk1/2, p-p38 MAPK, and NF-κB in the hippocampus of VTA-lesioned HFS PrL animals compared with VTA-lesioned sham animals. Studies have demonstrated that antidepressant drugs can enhance the protein expression of p-Akt to promote neurogenesis and neuroprotection against neuronal cell death [75]. It has been shown that an increase in p-Akt could inhibit p-GSK3-β via phosphorylation at its N terminus [76], and inhibit p-38 MAPK and NF-κB to reduce depressive-like behaviors by decreasing proinflammatory cytokines in an animal model of depression induced by repeated administration of lipopolysaccharide [73,74,77]. The DA receptors D2, D3, and D4 are robustly found in the hippocampus, cortical regions, striatum, nucleus accumbens and substantia nigra [78]. These receptors couple to G-proteins to inhibit adenylate cyclase and modulate Akt-GSK3 signaling to regulate neuronal differentiation and proliferation. Additionally, D1 and D2 receptors are known to modulate MAPK signaling, which in turn regulates neuronal plasticity and development, and cell death [79]. To further unravel the molecular mechanisms underlying the therapeutic effects of HFS PrL, we examined the regulatory pathway of Akt-GSK3 and caspase-3 activation on stress-induced apoptosis. As no differences were found in the levels of p-GSK3-β and caspase-3, we ruled out the possibility of Akt-GSK3 signaling mediating the antidepressant-like effects of HFS PrL in VTA-lesioned animals. 

In conclusion, animals exposed to prolonged stress can become susceptible or resilient to depressive symptoms and associated psychobiological pathologies. We demonstrated that stress-resilient HFS PrL animals had enhanced hedonia and reduced behavioral despair in the forced swim test together with reduced stress hormones when compared to sham animals. We demonstrated that HFS PrL has DA-dependent effects on anxiety and hedonic-like activities in both DRN- and VTA-lesioned animals. Compared with VTA-lesioned sham animals, VTA-lesioned HFS PrL animals showed decreased forced swim immobility, indicating the DA-independent effects were possibly mediated by hippocampal 5-HT neurotransmission and p-p38 MAPK/p-NF-κB mechanisms. Taken together, our findings showed that HFS PrL in stress-resilient animals induces profound antidepressant-like responses by enhancing hippocampal DA/5-HT neurotransmitters and reducing CORT stress hormone. These results suggest the stress-induced depressive-like behaviors induced by HFS PrL are mediated through the interactions of both DA-dependent and -independent mechanisms involving hippocampal neuroplasticity and neurotransmitter systems.

## Figures and Tables

**Figure 1 cells-12-01449-f001:**
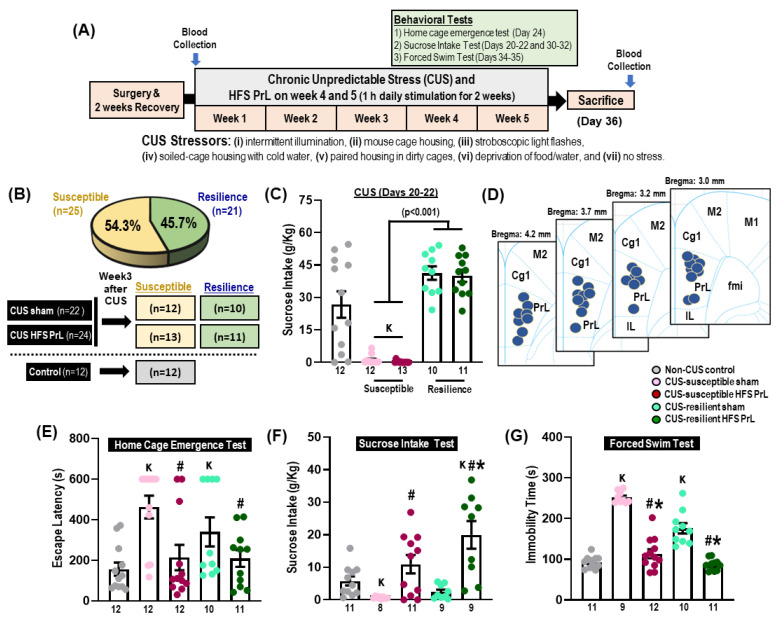
HFS PrL in CUS-resilient animals alleviates depressive-like behaviors. (**A**) Schematic representation of the experimental design of US paradigm exposure and behavioral testing. (**B**,**C**) Characterization of animals into susceptible and resilient groups based on their sucrose consumption level after 3 weeks of CUS exposure. (**D**) Localization of electrode tips within the PrL region. CUS induced significant reduction of sucrose consumption after 3 weeks stress exposure in susceptible groups (54.3%, *n* = 25) compared with resilient groups (H_(2)_ = 37.097, *p* < 0.001). (**E**) HFS PrL in both susceptible and resilient groups reduced anxiety-like behavior in the home cage emergence test compared with CUS-susceptible sham animals (H_(4)_ = 16.670, *p* = 0.002). (**F**,**G**) HFS PrL in CUS-resilient animals reduced anhedonic-like behavior (H_(4)_ = 22.142, *p* < 0.001) and forced swim immobility (H_(2)_ = 36.987, *p* < 0.001) compared with both CUS-susceptible and -resilient sham groups. Indication: K, significant difference from non-CUS controls, *p* < 0.05; **#**, significant difference from CUS-susceptible sham, *p* < 0.05; * significant difference from CUS-resilient sham, *p* < 0.05.

**Figure 2 cells-12-01449-f002:**
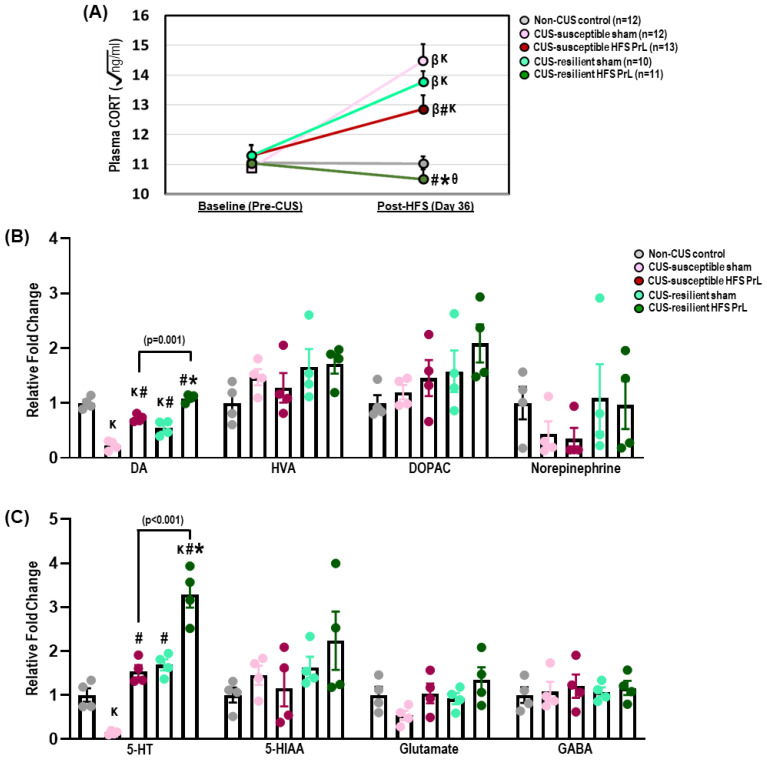
Effects of HFS PrL on the levels of corticosterone and hippocampal neurotransmitters. (**A**) HFS PrL in both CUS-susceptible and -resilient groups significantly reduced the levels of plasma CORT compared with their respective sham groups on day 36 of CUS (F_(4,53)_ = 16.908, *p* < 0.001). (**B**) Mass spectrometry analysis showed decreased DA in CUS-susceptible and -resilient sham groups compared with non-CUS controls (F_(4,15)_ = 52.892, *p* < 0.001). Interestingly, HFS PrL in both CUS-susceptible and -resilient animals increased DA levels compared with their respective sham groups. No significant differences were observed for HFS PrL on DA metabolites including HVA and DOPAC among groups (**B**,**C**). Although we found significant increases in 5-HT neurotransmitter levels in both CUS-susceptible and -resilient HFS PrL animals compared with their respective sham groups (F_(4,15)_ = 44.070, *p* < 0.001), there were no differences in 5-HIAA levels (**C**). No significant differences were observed for norepinephrine, glutamate, and GABA among groups. Indication: K, significant difference from non-CUS controls, *p* < 0.05; β, significant difference from baseline CORT before CUS procedures, *p* < 0.05; **#**, significant difference from CUS-susceptible sham, *p* < 0.05; *, significant difference from CUS-resilient sham, *p* < 0.05. θ, significant difference from CUS-susceptible HFS PrL, *p* < 0.05.

**Figure 3 cells-12-01449-f003:**
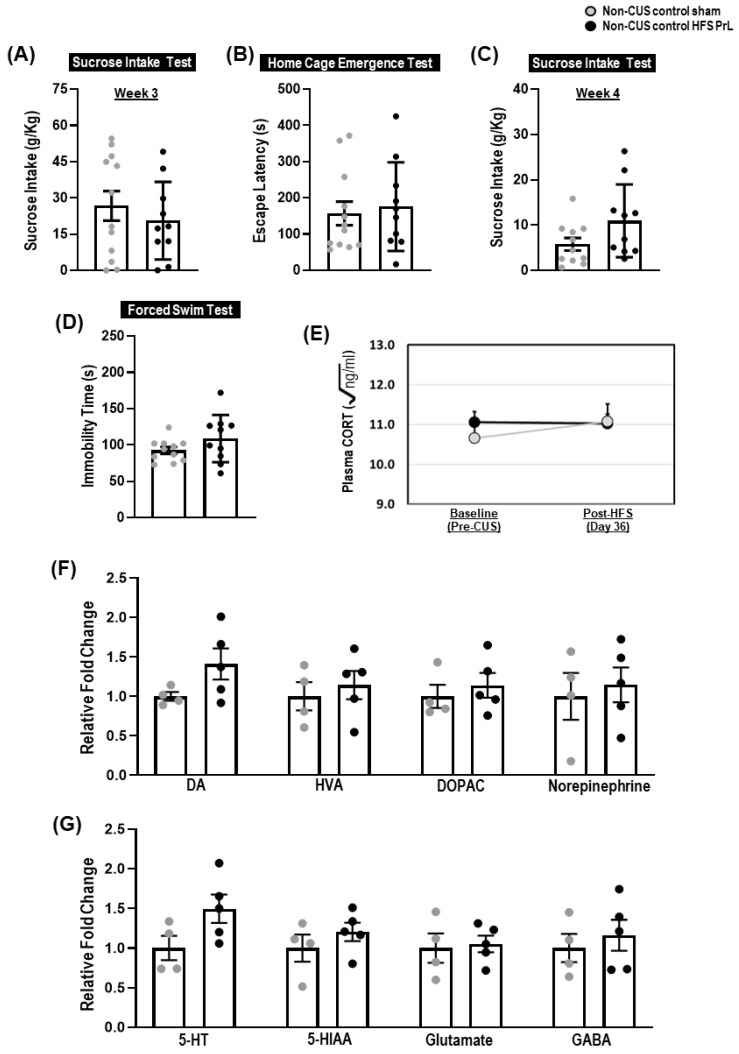
Effects of HFS PrL in non-CUS control animals. (**A**–**D**) No significant differences were found for sucrose intake test at week 3 (t_(20)_ = 0.751, *p* = 0.461), home cage emergency test (t_(20)_ = 1.088, *p* = 0.289), sucrose intake test at week 4 (t_(12_._350)_ = −1.987, *p* = 0.070), and forced swim immobility (t_(12_._470)_ = −1.445, *p* = 0.173) in non-CUS control HFS PrL compared with non-CUS control sham rats. (**E**) ANOVA with repeated-measures showed no significant changes of Time (F_(1,20)_ = 0.653, *p* = 0.429), Group (F_(1,20)_ = 0.209, *p* = 0.652) and their interactions (F_(1,20)_ = 0.902, *p* = 0.534) on plasma CORT levels between non-CUS control sham and HFS PrL animals. (**F**,**G**) No significant differences were found for DA (t_(7)_ = −1.802, *p* = 0.115), HVA (t_(6_._869)_ = −0.563, *p* = 0.591), DOPAC (t_(7)_ = −0.643, *p* = 0.541), Norepinephrine (t_(5_._895)_ = −0.395, *p* = 0.707), 5-HT (t_(7)_ = −2.049, *p* = 0.080), 5-HIAA (t_(7)_ = −1.021, *p* = 0.341), Glutamate (t_(7)_ = −0.256, *p* = 0.805), and GABA (t_(7)_ = −0.596, *p* = 0.570) in non-CUS control HFS PrL group compared with non-CUS control sham animals.

**Figure 4 cells-12-01449-f004:**
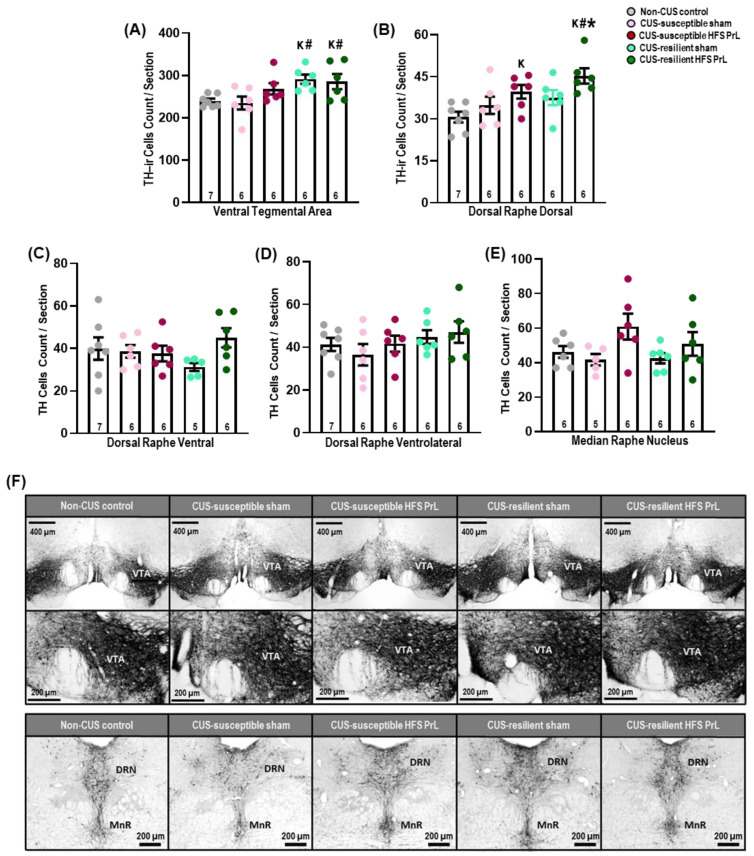
Effects of HFS PrL on midbrain TH-ir neurons in the CUS model. The bar graphs represent TH-ir cell count per section in the VTA (**A**), dorsal raphe dorsal (**B**), dorsal raphe ventral (**C**), dorsal raphe ventrolateral (**D**), and median raphe nucleus (**E**) of CUS and non-CUS control rats. (**F**) Representative low- and high-power photomicrographs of coronal sections in the VTA and DRN. (**A**) Interestingly, both CUS-resilient HFS PrL and sham animals showed increases in TH-ir cell count in the VTA compared with non-CUS control and CUS-susceptible sham groups (F_(4,26)_ = 3.934, *p* = 0.013). (**B**) In the dorsal raphe dorsal, animals with HFS PrL showed significantly increased TH-ir cell count in both CUS-susceptible and -resilient groups compared with non-CUS controls (F_(4,26)_ = 4.719, *p* = 0.005). (**C**–**E**) Statistical analysis test showed no significant differences were observed in the dorsal raphe ventral (F_(4,25)_ = 1.298, *p* = 0.298), dorsal raphe ventrolateral (F_(4,26)_ = 0.944, *p* = 0.454), and median raphe nucleus (F_(4,24)_ = 2.157, *p* = 0.105) of CUS and non-CUS control rats. Indication: K, significant difference from non-CUS controls, *p* < 0.05; **#**, significant difference from CUS-susceptible sham, *p* < 0.05. *, significant difference from CUS-resilient sham, *p* < 0.05.

**Figure 5 cells-12-01449-f005:**
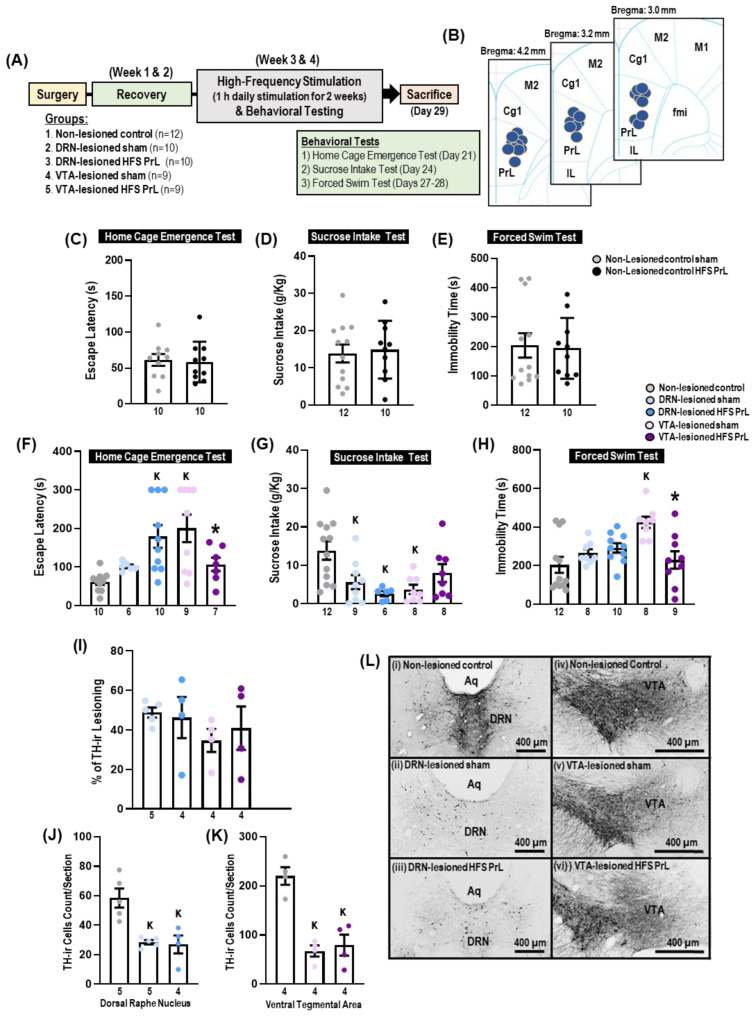
HFS PrL induces antidepressant-like effects through DA-dependent and -independent mechanisms. (**A**) Schematic representation of the experimental design for DA lesioning by micro-injection of 6-OHDA into the DRN and VTA, followed by behavioral testing. (**B**) Verification of electrode tips within the PrL region. (**C**–**E**) No significant differences were found for home cage emergency test (t_(18)_ = 0.234, *p* = 0.818), sucrose intake test (t_(20)_ = −0.152, *p* = 0.881), and forced swim immobility (Z = −0.264, *p* = 0.792) in non-lesioned control HFS PrL compared with non-lesioned control sham rats. (**F**) In the home cage emergence test, HFS PrL animals had reduced escape latency compared with VTA-lesioned sham animals (F_(2,23)_ = 9.727, *p* = 0.001). (**G**) In the sucrose intake test, ANOVA revealed remarkable reductions in sucrose consumption in DRN-lesioned sham and HFS PrL animals and VTA-lesioned sham animals compared with non-lesioned controls (F_(2,24–25)_ < 7.427, *p* < 0.010). VTA-lesioned HFS PrL animals showed no significant differences among the groups, indicating the anhedonic-like effects were dependent on DA function. (**H**) In the forced swim test, significantly increased immobility time was observed in VTA-lesioned sham animals, which was reversed with HFS PrL (F_(2,26)_ = 7.931, *p* = 0.002), indicating the antidepressant-like effects were independent of DA function. (**I**–**K**) The bar graphs represent the percentage of DA lesioning and TH-ir cell counts in the DRN and VTA of the rat model. (**L**) Representative photomicrographs of coronal sections of TH-ir staining in the DRN and VTA. (**I**–**K**) Animals with 6-OHDA lesioning of the DRN and VTA demonstrated significantly reduced TH-ir cell counts in the DRN (48.8%, DRN-lesioned sham; 46.3%, DRN-lesioned HFS PrL; F_(2,11)_ = 12.595, *p* = 0.001) and the VTA (34.7%, VTA-lesioned sham; 40.9%, VTA-lesioned HFS PrL; F_(2,9)_ = 23.889, *p* < 0.001), respectively. Indication: K, significant difference from non-lesioned controls, *p* < 0.05; *, significant difference from VTA-lesioned sham, *p* < 0.05.

**Figure 6 cells-12-01449-f006:**
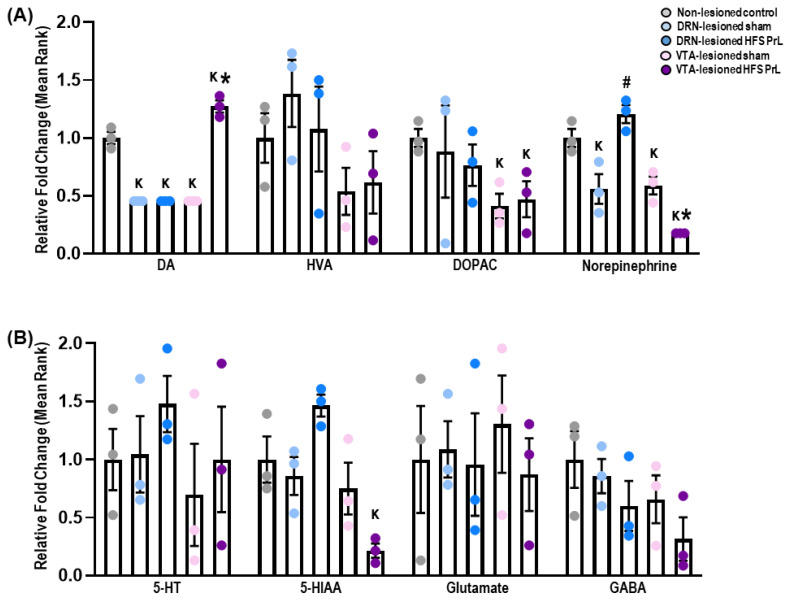
Effects of HFS PrL on the levels of hippocampal neurotransmission in DA-lesioned animals. (**A**) Mass spectrometry analysis demonstrated reduced levels of DA in DRN-lesioned sham and HFS PrL animals and VTA-lesioned sham animals compared with non-lesioned controls (all F_(2,6)_ < 108.000, *p* < 0.001). Interestingly, VTA-lesioned HFS PrL animals showed significantly increased DA and reduced norepinephrine levels compared with non-stimulated VTA-lesioned sham animals (all F_(2,6)_ < 94.500, *p* < 0.001). (**A**,**B**) There were no significant differences in HVA, 5-HT, glutamate, and GABA levels among the groups. Indication: K, significant difference from non-CUS controls, *p* < 0.05; **#**, significant difference from DRN-lesioned sham, *p* < 0.05; *, significant difference from VTA-lesioned sham, *p* < 0.05.

**Figure 7 cells-12-01449-f007:**
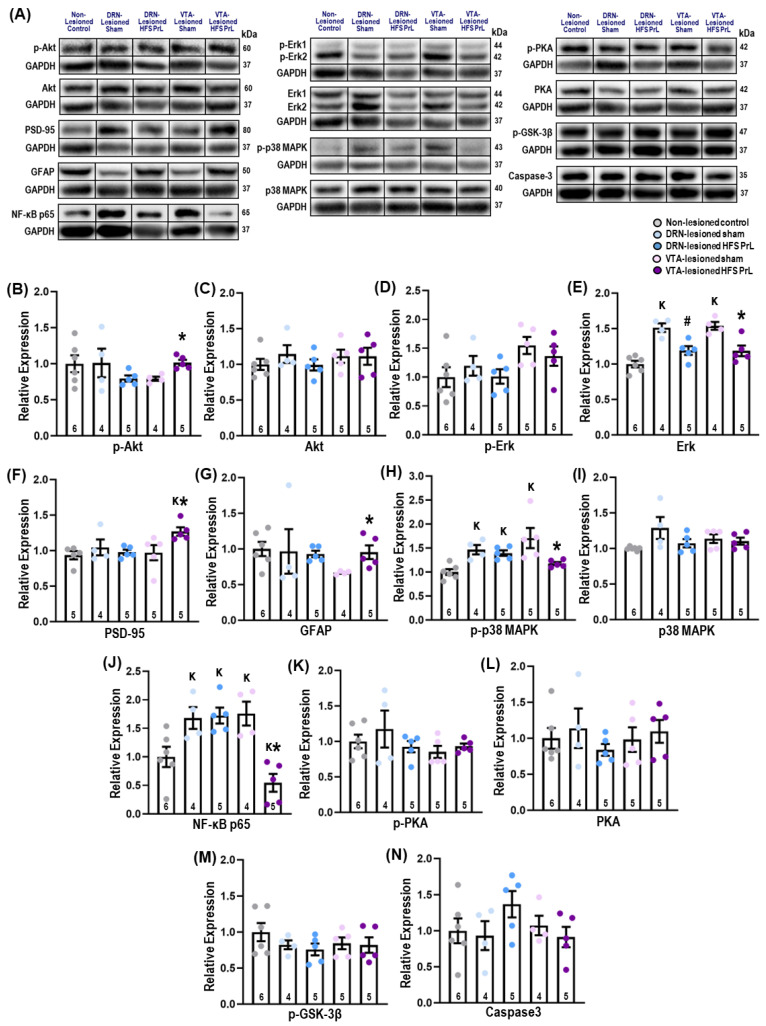
HFS PrL enhances hippocampal neuroplasticity-related function. (**A**) Graphical representation of the effects of HFS PrL on the protein expression of neuroplasticity-related markers. VTA-lesioned HFS PrL animals had significantly increased expressions of p-Akt (*p* = 0.003, (**B**)); PSD-95 (*p* = 0.026, (**F**)); and GFAP (*p* = 0.034, (**G**)) compared with VTA-lesioned sham group. Both VTA- and DRN-lesioned sham groups showed increased expression of Erk1/2, p-p38 MAPK, and NF-κB. Interestingly, VTA-lesioned HFS PrL animals had remarkably reduced expression of Erk1/2 (F_(2,12)_ = 19.890, *p* < 0.001, (**E**)); p-p38 MAPK (F_(2,13)_ = 9203, *p* = 0.003, (**H**)); and NF-κB (F_(2,12)_ = 10.038, *p* = 0.003, (**J**)) back to levels comparable to the non-lesioned controls. No significant differences were observed for Akt (**C**), p-Erk (**D**), p38 MAPK (**I**), p-PKA (**K**), PKA (**L**), p-GSK-3β (**M**), and caspase-3 (**N**) levels (all F_(2,12–13)_ < 3.473, *p* < 0.865) among the groups. Indication: K, significant difference from non-CUS controls, *p* < 0.05; **#**, significant difference from DRN-lesioned sham, *p* < 0.05; * significant difference from VTA-lesioned sham, *p* < 0.05.

## Data Availability

The data presented in this study are available on request from the corresponding author.

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
