# Peer review of "Prelimbic Cortical Stimulation Induces Antidepressant-like Responses through Dopaminergic-Dependent and -Independent Mechanisms"

_cells, 2023, doi:10.3390/cells12111449_

Round 1
Reviewer 1 Report
This manuscript explored the underlying mechanisms of HFS-induced antidepressant-like effect on susceptible and resilient rats. The overall idea is very clear. But when I review it, I found a little difficult to understand some content.
1. As you mentioned several times about treatment-resistant depression, this gave me a misleading that you did some research about the effect and mechanism of HFS on treatment-resistant depression. However, they mainly focused on susceptible and resilient rats.
2. In method part, as shown in Fig. 1A, "HFS PrL on week 4 and 5", what this mean? It means HFS was given two weeks or only 15 min before each test and the entire test phase except the forced swim (Line 127-130)? Besides, why didn't you do these behavioral tests all around Day 35?
3. In Fig. 1C, the sucrose intake for all groups are very different from the results in other Figures, like Fig. 1F? Besides, in Fig.1F, CUS-resilient sham group also showed very low sucrose intake, which looks like CUS-susceptible sham. Can you explain?
4. In result part, you found decreased DA levels, so what kind of tissue did you use? And how did you count the number of TH-ir cells?
5. In the statement of result section, some conclusion needs to have stronger support, for example, in Line 300-302, if you want to know whether HFS PrL induced anxiety-like responses, you should compare DRN-lesioned HFS PrL with DRN-lesioned sham rather than non-lesioned controls. Besides, I have a suggestion about avoiding unnecessary comparation. For example, in Line 205-207, I don't think comparing CUS-resilient HFS PrL with CUS-susceptible sham is necessary.
6. Why did you choose hippocampus for DA neurotransmission and WB? I know DA neurons have projection to stratum?
7. For experiment 2, you used DRN/ VTA-lesion to build another depression rat model with damaged DA neurons. That's good. A clear explanation can make these two experiments have a closer connection. But in WB blot part, as the protein levels in DRN-lesioned sham and VTA-lesioned sham has no significant changes compared with Non-lesioned control, it's hard to say HFS has anti-depressant effect through pAkt-PSD-95 pathway.
8. In Fig. 4D, with DRN-lesion, HFS couldn't rescue the decrease sucrose intake, which means DA neurons in DRN are involved in HFS's anti-depression. However, VTA-lesion seems to have less effect on HFS's anti-depression.
The English needs to be improved.
Author Response
Reply to Comments of Reviewer 1
Comment: This manuscript explored the underlying mechanisms of HFS-induced antidepressant-like effect on susceptible and resilient rats. The overall idea is very clear. But when I review it, I found a little difficult to understand some content.
- As you mentioned several times about treatment-resistant depression, this gave me a misleading that you did some research about the effect and mechanism of HFS on treatment-resistant depression. However, they mainly focused on susceptible and resilient rats.
Reply: We appreciate the comment of the reviewer, the word ‘treatment-resistant depression’ has been removed from the manuscript.
Comment: 2. In method part, as shown in Fig. 1A, "HFS PrL on week 4 and 5", what this mean? It means HFS was given two weeks or only 15 min before each test and the entire test phase except the forced swim (Line 127-130)? Besides, why didn't you do these behavioral tests all around Day 35?
Reply: We apologize for the confusion, and we have indicated “HFS PrL on week 4 and 5 (1h daily stimulation for 2 weeks)” in Fig. 1A and also in the material and methods section. In all behavioral tasks, animals were stimulated for 15 min prior to each test with continual stimulation during the entire testing phase, except the forced swim test received stimulation only prior to testing. The reason that we did not perform all the behavioural tests at around day 35, because we would like to ensure that there was a sufficient washout period to avoid the behavioural carry-over effects for each test.
Comment: 3. In Fig. 1C, the sucrose intake for all groups are very different from the results in other Figures, like Fig. 1F? Besides, in Fig.1F, CUS-resilient sham group also showed very low sucrose intake, which looks like CUS-susceptible sham. Can you explain?
Reply: In Fig. 1C, the sucrose intake test was conducted after 3 weeks of CUS (on days 20-22). The results are different from Fig. 1F, because in Fig. 1C, animals were categorized as either susceptible or resilient to CUS-induced depression based on the sucrose intake test after week 3 of CUS. In accordance with earlier studies, animals were categorized as susceptible to depressive-like or anhedonia-like behavior if there was a 40% reduction in sucrose consumption compared with the average sucrose intake of the non-CUS controls [Bergstrom et al, 2007; Bergstrom et al, 2008], whereas the remaining animals were categorized as resilient to depressive-like behavior (Fig. 1B-C).
In Fig. 1F, the CUS-resilient sham group showed very low sucrose intake, it is most likely that these animals developed anhedonic-like response at week 4 (on day 30-32). Nevertheless, the CUS-resilient sham group showed no statistical difference compared to the control, while animals with HFS of both the CUS-susceptible and -resilient groups enhanced the hedonic-like responses compared with sham, respectively.
References:
- Bergstrom A, Jayatissa MN, Thykjaer T, Wiborg O. Molecular pathways associated with stress resilience and drug resistance in the chronic mild stress rat model of depression: a gene expression study. J Mol Neurosci. 2007;33(2):201-15.
- Bergstrom A, Jayatissa MN, Mork A, Wiborg O. Stress sensitivity and resilience in the chronic mild stress rat model of depression; an in situ hybridization study. Brain Res. 2008;1196:41-52.
Comment: 4. In result part, you found decreased DA levels, so what kind of tissue did you use? And how did you count the number of TH-ir cells?
Reply: In the result part of Fig. 2B-C, the hippocampus was micro-dissected for measurement of changes of neurotransmitters and their metabolites.
The quantification of TH-ir cells was performed in the VTA (Bregma level: from -5.0 to -6.0 mm) and different regions of the DRN (including dorsal raphe dorsal, dorsal raphe ventral, dorsal raphe ventrolateral, and median raphe nucleus; Bregma level: from −7.3 to −8.2 mm), as previously described [Tan et al, 2020 Brain Struct Funct.].]. Photomicrographs of TH-ir cells within the regions of interest (4-5 sections per animal) were taken using an Olympus DP73 digital camera (Olympus, Hamburg, Germany) attached to an Axiophat 2 imaging microscope (Carl Zeiss Microscopy GmbH, Gottingen, Germany) and quantification was performed using ‘Image J’ (version 1.38, NIH, USA) as previously described [Tan et al, 2020 Brain Struct Funct.]. In-section artefacts were excluded from the analysis to ensure accuracy of the measurements.
We have included this description in the manuscript, see lines 242-250.
Comment: 5. In the statement of result section, some conclusion needs to have stronger support, for example, in Line 300-302, if you want to know whether HFS PrL induced anxiety-like responses, you should compare DRN-lesioned HFS PrL with DRN-lesioned sham rather than non-lesioned controls. Besides, I have a suggestion about avoiding unnecessary comparation. For example, in Line 205-207, I don't think comparing CUS-resilient HFS PrL with CUS-susceptible sham is necessary.
Reply: We agreed with the comments of the reviewer. We compared only the effect of DRN lesioned HFS PrL with the DRN lesioned sham rather than with the non-lesioned controls. Additionally, we have removed all the unnecessary comparisons, accordingly.
Comment: 6. Why did you choose hippocampus for DA neurotransmission and WB? I know DA neurons have projection to stratum?
Reply: The reason that the hippocampus was selected for measurement of neurotransmission and Western blot study due to the projection of VTA to the hippocampus (Tsetsenis et al, 2021; Han et al, 2020). The mesolimbic system provides DA innervation to subcortical regions including the NAc, septum, olfactory tubercle, hippocampus, and amygdala. Many studies have now started focusing on this mesohippocampal pathway, but the synaptic properties of these projections have only been characterized for VTA DA neurons (Rocchetti et al., 2015 Biol Psychiatry; Rosen et al., 2015 Nat Neuroscience). Therefore, we hypothesized that these neurons might constitute an additional functional connection to the hippocampus.
References:
- Tsetsenis T, Badyna JK, Wilson JA, Zhang X, Krizman EN, Subramaniyan M, et al. Midbrain dopaminergic innervation of the hippocampus is sufficient to modulate formation of aversive memories. Proc Natl Acad Sci U S A. 2021;118(40).
- Han Y, Zhang Y, Kim H, Grayson VS, Jovasevic V, Ren W, et al. Excitatory VTA to DH projections provide a valence signal to memory circuits. Nat Commun. 2020;11(1):1466.
- Rocchetti et al. Presynaptic D2 dopamine receptors control long-term depression expression and memory processes in the temporal hippocampus. Biol Psychiatry. 2015 Mar 15;77(6):513-25.
- Rosen et al. Midbrain dopamine neurons bidirectionally regulate CA3-CA1 synaptic drive. Nat Neurosci. 2015 Dec;18(12):1763-71.
Comment: 7. For experiment 2, you used DRN/ VTA-lesion to build another depression rat model with damaged DA neurons. That's good. A clear explanation can make these two experiments have a closer connection. But in WB blot part, as the protein levels in DRN-lesioned sham and VTA-lesioned sham has no significant changes compared with Non-lesioned control, it's hard to say HFS has anti-depressant effect through pAkt-PSD-95 pathway.
Reply: We thank the reviewer for the suggestion and we have provided a description to explain the connection of experiments 1 and 2. See Material & Methods under “Experiment 2: HFS PrL in animal models with 6-OHDA lesioning in DRN and VTA”.
We agreed with the reviewer that it’s hard to conclude that the antidepressant-like effects induced by HFS were mediated through the pAkt-PSD-95 pathway, we have modified this statement accordingly in the abstract and manuscript.
Comment: 8. In Fig. 4D, with DRN-lesion, HFS couldn't rescue the decrease sucrose intake, which means DA neurons in DRN are involved in HFS's anti-depression. However, VTA-lesion seems to have less effect on HFS's anti-depression.
Reply: Indeed, we agreed with the reviewer that no rescue was observed with HFS in DRN-lesioned group and this could mean that the DA neuron in DRN were involved in the effects of HFS on anti-anhedonic-like activities. Whereas for VTA lesioned, although we did not observe an increase of sucrose intake after HFS compared to sham animals, there was also no significant difference compared to the non-lesioned control group, possibly indicates the reversal from anhedonia condition to a normal hedonic level.

Reviewer 2 Report
Khairuddin and colleagues report that prelimbic cortical stimulation induces antidepressant-like responses through dopaminergic-dependent and -independent mechanisms. Overall, this is an interesting study mostly because the Authors segregated rats in susceptible and resilient subpopulations. Despite these findings would be of general interest to this field of research, the work raises concerns that need to be addressed.
Major points
- It is questionable the design of experiment 2. Whereas the experiment 1 was performed by using the CUS model, which is well-established model to study depression, the experiment 2 was performed on not stressed rats. Despite the results obtained suggest an antidepressant-like effect of HFS Prl that depends on dopaminergic mechanisms, it would be more translational and interesting to understand what can occur in DRN- or VTA-lesioned stress susceptible rats. The Authors should at least clarify why they did not use the CUS model in experiment 2.
- The discussion of the results is almost absent. The Authors must avoid to report in the discussion again the experimental design and the results obtained. The Authors must only discuss the results obtained according to the available appropriate literature.
Minor points
- It is quite strange that the Authors mentioned in the abstract mostly CUS resilient rats. The Authors should emphasize the beneficial effect of HFS Prl on CUS susceptible rats, which exhibits phenotypes related to depression.
- I suggest to better discuss the involvement of the DA system in the pathophysiological mechanisms underlying not only depression but in general trauma and stressor related disorders. I suggest to add (line 57…) and discuss more recent papers describing the involvement of the DA system in the pathophysiological mechanisms of trauma and stressor related disorders (PMID: 36400332; 34253715; 29106542) .
- The material and methods section should be better organized with subparagraphs.
- Line 116: “animals were categorized as susceptible to depression”. This is not correct. Depression is a human pathology. Rats or mice can exhibits depressive-like phenotypes.
- Some results are unclear. Please rephrase lines 237-239.
Minor editing of English language is required.
Author Response
Reply to Comments of Reviewer 2
Comment: Khairuddin and colleagues report that prelimbic cortical stimulation induces antidepressant-like responses through dopaminergic-dependent and -independent mechanisms. Overall, this is an interesting study mostly because the Authors segregated rats in susceptible and resilient subpopulations. Despite these findings would be of general interest to this field of research, the work raises concerns that need to be addressed.
Major points
Comment: It is questionable the design of experiment 2. Whereas the experiment 1 was performed by using the CUS model, which is well-established model to study depression, the experiment 2 was performed on not stressed rats. Despite the results obtained suggest an antidepressant-like effect of HFS Prl that depends on dopaminergic mechanisms, it would be more translational and interesting to understand what can occur in DRN- or VTA-lesioned stress susceptible rats. The Authors should at least clarify why they did not use the CUS model in experiment 2.
Reply: We agreed with the reviewer that it would be more appropriate to conduct the DRN- or VTA-lesioning in stress susceptible rats. Nevertheless, as we have shown a remarkable reduction of dopamine level in the CUS-susceptible and CUS-resilient sham compared to non-CUS control, our experiment 2 aimed to investigate whether the lesioning of dopamine alone in the DRN or VTA would produce similar effects of depressive-like behaviours as the CUS model. We have demonstrated that DRN-lesioned animals induced anhedonic response, while VTA-lesioned animals showed anxiety (increased escape latency in the home cage emergence test), anhedonic (decreased sucrose intake), and increased forced swim immobility activities, these behavioural phenotypes are similar to the model induced by CUS paradigm.
Comment: The discussion of the results is almost absent. The Authors must avoid to report in the discussion again the experimental design and the results obtained. The Authors must only discuss the results obtained according to the available appropriate literature.
Reply: We agreed with the comment of the reviewer, parts of the discussion have been rewritten accordingly.
Minor points
Comment: It is quite strange that the Authors mentioned in the abstract mostly CUS resilient rats. The Authors should emphasize the beneficial effect of HFS Prl on CUS susceptible rats, which exhibits phenotypes related to depression.
Reply: We appreciate the comment of the reviewer, both the CUS-susceptible and resilient groups have been included in the abstract.
Comment: I suggest to better discuss the involvement of the DA system in the pathophysiological mechanisms underlying not only depression but in general trauma and stressor related disorders. I suggest to add (line 57…) and discuss more recent papers describing the involvement of the DA system in the pathophysiological mechanisms of trauma and stressor related disorders (PMID: 36400332; 34253715; 29106542).
Reply: We appreciate the comment of the reviewer, additional discussion regarding the involvement of DA system in the pathophysiological mechanisms of trauma and stressor related disorder has been added into the discussion section.
Comment: The material and methods section should be better organized with subparagraphs.
Reply: We thank the suggestion of the reviewer; the material and methods have been reorganized into subparagraphs.
Comment: Line 116: “animals were categorized as susceptible to depression”. This is not correct. Depression is a human pathology. Rats or mice can exhibits depressive-like phenotypes.
Reply: We agreed with the comment of the reviewer, the “depression” has been changed to “depressive-like behavior”
Comment: Some results are unclear. Please rephrase lines 237-239.
Reply: We have rephrased the sentence from “Interestingly, 1 h of HFS PrL in both CUS-susceptible and -resilient groups reduced CORT levels back to the respective levels in the CUS sham groups (p<0.044) on day 36 of CUS before animals were sacrificed.” to the following “On day 36 of CUS before animals were sacrifice, HFS PrL in both CUS-susceptible and -resilient groups significantly reduced CORT levels compared to their respective CUS sham groups (p<0.044).”

Round 2
Reviewer 1 Report
The current manuscript looks good. They answered all my questions carefully and with evidence.
Reviewer 2 Report
The Authors addressed all the points I raised.
Well done.